# Isogenic Mammary Models of Intraductal Carcinoma Reveal Progression to Invasiveness in the Absence of a Non-Obligatory In Situ Stage

**DOI:** 10.3390/cancers15082257

**Published:** 2023-04-12

**Authors:** Sarah M. Bernhardt, Elizabeth Mitchell, Stephanie Stamnes, Reuben J. Hoffmann, Andrea Calhoun, Alex Klug, Tanya D. Russell, Nathan D. Pennock, Joshua M. Walker, Pepper Schedin

**Affiliations:** 1Department of Cell, Developmental and Cancer Biology, Oregon Health & Science University, Portland, OR 97239, USA; 2Center for Advancing Professional Excellence, University of Colorado Anschutz Medical Campus, Aurora, CO 80045, USA; 3Department of Radiation Medicine, Oregon Health & Science University, Portland, OR 97239, USA; 4Knight Cancer Institute, Oregon Health & Science University, Portland, OR 97239, USA; 5Young Women’s Breast Cancer Translational Program, University of Colorado Cancer Center, Aurora, CO 80045, USA

**Keywords:** immune-competent, mouse-intraductal (MIND) model, myoepithelium, pure invasive ductal carcinoma, ductal carcinoma in situ

## Abstract

**Simple Summary:**

Breast cancer—ductal carcinoma of the breast and invasive ductal carcinoma (IDC)—arises within the confines of the mammary ductal epithelium. Progression to IDC can occur through a pre-invasive, ductal carcinoma in situ stage (DCIS), or in the absence of DCIS. The immune system has recently been identified as a factor in disease progression, highlighting the need for immune-competent mouse models of pre-invasive disease. To model early-stage disease, we tested six distinct murine mammary tumor cell lines injected directly into the mammary ducts of immune-competent mice. We find that all six mouse cell lines bypassed a stable DCIS stage, rapidly progressing to IDC. Similarly, in immune-compromised mice, we observed IDC in the absence of DCIS, suggesting an intact immune system may not play a primary role in early disease progression in these mouse models. These models may be useful for the study of IDC that occur in the absence of DCIS, and in the development of immune therapies.

**Abstract:**

In breast cancer, progression to invasive ductal carcinoma (IDC) involves interactions between immune, myoepithelial, and tumor cells. Development of IDC can proceed through ductal carcinoma in situ (DCIS), a non-obligate, non-invasive stage, or IDC can develop without evidence of DCIS and these cases associate with poorer prognosis. Tractable, immune-competent mouse models are needed to help delineate distinct mechanisms of local tumor cell invasion and prognostic implications. To address these gaps, we delivered murine mammary carcinoma cell lines directly into the main mammary lactiferous duct of immune-competent mice. Using two strains of immune-competent mice (BALB/c, C57BL/6), one immune-compromised (severe combined immunodeficiency; SCID) C57BL/6 strain, and six different murine mammary cancer cell lines (D2.OR, D2A1, 4T1, EMT6, EO771, Py230), we found early loss of ductal myoepithelial cell differentiation markers p63, α-smooth muscle actin, and calponin, and rapid formation of IDC in the absence of DCIS. Rapid IDC formation also occurred in the absence of adaptive immunity. Combined, these studies demonstrate that loss of myoepithelial barrier function does not require an intact immune system, and suggest that these isogenic murine models may prove a useful tool to study IDC in the absence of a non-obligatory DCIS stage—an under-investigated subset of poor prognostic human breast cancer.

## 1. Introduction

In breast cancer, invasive ductal carcinoma (IDC) is characterized by malignant cancer cells breaching the myoepithelial cell layer and escaping the confines of the mammary duct. There are distinct patterns of invasive disease, suggesting different mechanisms of early-stage disease progression. IDC is frequently diagnosed concurrent with ductal carcinoma in situ (DCIS), implicating progression through an in situ stage, whereby tumor cells are confined within mammary ducts prior to the transition to local invasion. By contrast, pure IDCs are invasive carcinomas that, at the time of diagnosis, do not have a clinically identifiable DCIS component, and are thought to progress in the absence of a DCIS precursor [1,2]. There is significant variance in the literature regarding the prevalence of pure IDC compared to IDC with concurrent DCIS, with pure IDC ranging from 23.1% to 78.8% of all IDC cases [3,4,5,6,7,8]. Pure IDC has been reported as more aggressive than IDC concurrent with DCIS [3,9,10], presenting as larger, higher-grade tumors that are more proliferative and frequently ER-negative, with a shorter overall survival [8,9,10]. The mechanisms involved in early-stage invasive disease, in the presence or absence of a DCIS stage, are not well understood.

In early-stage disease, the myoepithelium is thought to serve as an active barrier to inhibit tumor cell invasion [11,12,13,14,15,16]. Differentiated myoepithelial cells are characterized by the expression of p63, a member of the p53 family of transcription factors and biomarker of basal phenotype, α-smooth muscle actin (SMA), and calponin, an actin-binding protein that mediates smooth muscle function [17,18,19]. In a xenograft mouse model of human breast cancer progression, myoepithelial cells lose expression of these differentiation markers prior to local tumor invasion [20]. In human pre-invasive disease, a similar loss in myoepithelial cell differentiation markers is observed [19,20]. Cumulatively, these studies support a role for de-differentiation of myoepithelial cells in progression to IDC. Importantly, loss of myoepithelial differentiation is observed in pure IDC, as well as IDC with concurrent DCIS, suggesting that myoepithelial de-differentiation is a key event in the progression to invasive disease.

Several recent studies have associated focally compromised myoepithelial cells with immune cell infiltration and T-cell activation [19,21], including a recent report associating physical gaps in the myoepithelium with anti-tumor immunity and reduced disease recurrence [22]. Combined, these studies suggest divergent roles for adaptive immunity in the transition from DCIS to IDC: a pro-tumor role, in which T cells may contribute to the loss of myoepithelial barrier function, and an anti-tumor role, in which T cells are specifically recruited to the site of compromised myoepithelium, where they may provide early tumor control. This ambiguous association observed in both human specimens and mouse models highlights the need for additional studies evaluating the role of immune cells in myoepithelial cell integrity and early-stage disease progression.

Currently, there is a lack of isogenic, immune-competent mouse models of breast cancer progression. The most common tractable mouse models of breast cancer are xenograft models, where patient-derived breast cancer cell lines or breast tumor explants are orthotopically injected into the mammary fat pad of immunocompromised mice. While critically important research tools, these mammary fat pad models bypass the biological processes of tumor cell escape from the confines of the mammary ducts and lack an intact immune system. Transgenic mice can meet both of these model requisites, as they exhibit breast cancer progression from intraductal pre-invasive to invasive stages in immune-competent hosts. In these models, tissue-specific mutations, commonly found in human breast cancer [23,24], are expressed under mammary-specific, hormone-responsive promoters resulting in broad oncogene expression in hormone-responsive mammary epithelial cells. These mice develop florid intraductal hyperplasia that heterogeneously progresses to multifocal invasive disease. Thus, due to heterogeneity in tumor development, transgenic mouse models are also not well suited to study early-stage disease in isolation.

Mouse mammary intraductal (MIND) models represent anatomically correct models for the study of pre-invasive disease [20,25,26,27,28]. Here, human breast cancer cells are injected directly into the main lactiferous mammary duct of mice. MIND models permit assessment of early interactions between intraductal cancer cells, normal mammary epithelium, and the myoepithelium, as well as the underlying basement membrane. Previous studies using human breast cancer cell lines demonstrate that intraductal injection results in human tumor cell incorporation into the ductal epithelium [20], followed by the development of robust DCIS-like lesions with relatively long latency to invasive cancer [25,29,30,31]. Of note, these MIND models most frequently use invasive human breast cancer cell lines as opposed to pre-cancerous cell lines, yet robustly develop stable DCIS lesions. These observations are consistent with previous studies demonstrating that DCIS and IDC are molecularly indistinguishable, with the majority of tumor-associated oncogenic changes occurring at the transition from normal to DCIS [32,33]. At the transition from DCIS to IDC in human breast cancer, transcriptomic changes associate primarily with cells of the tumor microenvironment, including the myoepithelium and immune milieu [32,33]. Thus, to further our understanding of how the tumor microenvironment contributes to early-stage disease progression, we sought to develop immune-competent mouse models of intraductal-stage disease.

Here, we combined a pre-established intraductal tumor cell delivery model with isogenic murine mammary carcinoma cell lines and immune-competent hosts. We used two different immune-competent mouse strains, BALB/c and C57BL/6, and one immune-compromised strain (severe combined immunodeficient mice; SCID), six different mammary tumor cell lines (D2A1, D2.OR, 4T1, EMT6, EO771, Py230), and three human breast cancer lines (HCC70, MCF7, T47D). We find that intraductal delivery of mouse mammary tumor cells resulted in compromised myoepithelium and the development of IDC, with little evidence of a stable DCIS stage, whereas the human breast cancer cells developed robust DCIS-like lesions. To determine if rapid progression to IDC was associated with a fully intact immune compartment in these isogenic mouse models, we utilized SCID hosts and demonstrate that rapid progression to IDC was not dependent on adaptive immunity. We suggest that intraductal injection of these mouse mammary tumor cells may model human pure IDC disease, which develops in the absence of a discernable DCIS precursor stage.

## 2. Materials & Methods

### 2.1. Cell Culture

Six different murine mammary carcinoma cell lines were used in this study. Four BALB/c cell lines that were derived from three independent mammary tumors established in the BALB/c mouse strain (4T1, D2A1, D2.OR, EMT6); and two C57BL/6 mammary tumor cell lines that were independently derived from mammary tumors established in the C57BL/6 mouse strain (Py230, EO771). These cell lines have different intrinsic subtypes [34,35,36] (Appendix A). The 4T1 cell line was provided by Dr. Heide Ford (University of Colorado, Aurora, CO, USA), and cultured in DMEM, supplemented with 10% fetal calf serum (FCS), 1% L-Glutamine, and 1% MEM non-essential amino acids. D2A1 and D2.OR cell lines were generously donated by Jeffrey E. Green (Laboratory of Cell Biology and Genetics, National Cancer Institute), and cultured in DMEM supplemented with 10% FCS. EMT6 cells were purchased from American Type Culture Collection (ATCC) and cultured in DMEM supplemented with 10% FCS. Py230 cells were purchased from ATCC and cultured in Hams F-12K Medium, supplemented with 0.1% MITO+ serum extender, and 5% FCS. EO771 cells were purchased from CH3 Biosystems, and cultured in RPMI1640 supplemented with 5% FCS.

Three different human breast cancer cell lines were used: MCF7, T47D, and HCC70. All cell lines were obtained from the University of Colorado Cancer Center Protein Production/Mab/Tissue Culture Core and cultured as previously described [20].

### 2.2. Tumor Cell Motility Assay

Murine cancer cell lines (4T1, D2A1, D2.OR, EMT6, Py230, EO771) and human breast cancer cell lines (MCF7, T47D, HCC70) were plated in technical triplicate in six-well plates (5 × 10^5^ cells/well, 2 mL), under the media conditions described above. After reaching confluency, cells were starved overnight in serum-free media. Cell monolayers were scratched using a p200 pipette tip prior to being washed twice with PBS, and 2% serum media was added. Three consistent regions of interest were imaged per well at 0 h, 4 h, 8 h, and 24 h timepoints, using a Nikon Eclipse Ts2R microscope and accompanying DS-Qi2 camera. The rate of wound healing was assessed by measuring and averaging the distance between the borders of the wound at nine evenly-distributed regions per replicate, using Image J (Fiji). Data were normalized to the 0 h time point. Experiments were repeated 3 to 5 times, and data were averaged and presented as mean distance of closure ± standard error of the mean (SEM).

### 2.3. Human Tissue Sample Collection

Formalin-fixed, paraffin-embedded (FFPE) invasive breast cancer samples were obtained under an Oregon Health & Science University (OHSU) Institutional Review Board protocol. All cases were de-identified to the research team at all points.

### 2.4. Animal Studies

All animal procedures were approved by the OHSU Institutional Animal Care and Use Committee. Female BALB/c mice of 9–12 weeks old and female C57BL/6 mice of 10 weeks old were purchased from Charles River Laboratories (Wilmington, MA, USA). For experiments using severe combined immunodeficient mice (SCID) animals, 10-week-old B6.CB17-Prkdcscid/SzJ were purchased from The Jackson Laboratory (Bath Harbor, MA, USA).

### 2.5. Mouse Intraductal Injection

For the intraductal delivery model, tumor cells were injected into the mammary teat in the absence of any surgical manipulation, as previously described [20,37]. In brief, mice were anesthetized with isoflurane and the hair surrounding the third thoracic and fourth inguinal mammary glands trimmed with fine scissors and wiped with 70% ethanol. A 25-μL Wiretrol II disposable glass micropipette (no. 5-000-2050; Drummond Scientific Company, Broomall, PA, USA) was drawn and fire-polished into a fine tip of 60–75 μm. The glass pipette was washed with 70% ethanol, then rinsed with 1 × PBS. Tumor cells suspended in PBS were back-loaded into the micropipette using a stainless-steel plunger. The micropipette tip was gently inserted directly into the teat canal with the help of a micromanipulator, and tumor cells were slowly ejected into the mammary glands of mice. Mammary cancer cell lines were injected at various concentrations: D2A1 (1 × 10^4^ cells or 5 × 10^4^ cells), D2.OR (1 × 10^4^ cells or 5 × 10^4^ cells), EMT6 (1 × 10^4^ cells or 5 × 10^4^ cells), 4T1 (1 × 10^4^ cells or 5 × 10^4^ cells), Py230 (1 × 10^4^ cells or 5 × 10^4^ cells), and EO771 (5 × 10^3^ cells or 5 × 10^4^ cells) depending on the experiment. Cell concentrations were selected based on previously published data [20,25,27,29], or unpublished data from our lab group. An overview of the intraductal delivery method is presented in Figure 1A. To avoid physical disruption to the mammary ducts, cells were injected in a total volume of 2 µL, except for experiments in Figure 1B,C, which used a 5 µL injection volume, as previously described [20].

### 2.6. Mammary Fat Pad Injections

Mice were anesthetized with isoflurane, and 10 µL of murine mammary cancer cell lines EO771 (5 × 10^4^ in 10 µL PBS) or Py230 (5 × 10^4^ in 10 µL PBS) were injected into the 4th mammary fat pad of C57BL/6 mice, using an insulin syringe (1 mL, 28 G).

### 2.7. Mammary Whole Mount Staining

Inguinal mammary glands were dissected from mice and stained as whole mounts with carmine aluminum. Briefly, mammary glands were air-dried on glass slides for 5 min, then fixed in modified Carnoy’s fixative (25% glacial acetic acid, 75% ethanol) for 2 h at room temperature. Mammary glands were transferred to carmine aluminum stain (0.2% carmine, 0.5% aluminum potassium sulfate) overnight at room temperature with agitation. Slides were dehydrated by passing through a series of increasing ethanol concentrations, cleared with xylene, and mounted using Cytoseal 60. Whole mount images were captured using a Zeiss Axio Observer Z1 Microscope, and images were analyzed using Zeiss ZEN software.

Following whole mount analysis, mammary glands were processed for histological and immunohistochemical analysis. Briefly, coverslips were removed from whole mounts following imaging by soaking in xylene for 6 h. Mammary glands were transferred into tissue cassettes prior to embedding in paraffin wax for subsequent thin sectioning.

### 2.8. Hematoxylin–Eosin Staining and Immunohistochemistry

Hematoxylin and eosin (H&E) and immunohistochemical (IHC) staining were performed on 4µm thick tissue sections. For H&E analysis of human breast cancer samples and murine mammary glands and tumors, sections were dewaxed in xylene and subsequently passed through decreasing ethanol concentrations for rehydration. Slides were stained with hematoxylin and counterstained with eosin prior to dehydrating and mounting with mounting medium.

For IHC analysis of murine mammary glands and tumors, sections were dewaxed in xylene, and passed sequentially through decreasing concentrations of ethanol for rehydration. Antigen retrieval was performed in EDTA solution (Dako, Agilent, Santa Clara, CA, USA) in a pressure cooker (Dako Pascal, Agilent, Santa Clara, CA, USA) at 125 °C for 5 min. Slides were blocked for endogenous peroxidases using 3% H_2_O_2_ in methanol. Slides were subsequently incubated in 5% normal goat serum and 2.5% bovine serum albumin to block non-specific binding sites. Sections were incubated with primary antibodies, including p63 (BioCare, Pacheco, CA, USA, 1:200, 1 h at room temperature), smooth muscle actin (SMA) (Abcam, Cambridge, UK, 1:400, 1 h at room temperature), calponin (Abcam, 1:4000, 1 h at room temperature), and CD45 (BD Pharm, San Diego, CA, USA, 1:50, 1 h at room temperature). Slides were washed in 1 × TBST prior to incubation with a pre-diluted HRP-conjugated secondary antibody (Dako). Antibody binding was visualized using AEC (HRP; Vector Laboratories, Newark, CA, USA) or DAB (HRP; Agilent, Santa Clara, CA, USA) peroxidase, as per manufacturer’s instructions. Slides were counterstained with hematoxylin prior to dehydrating and coverslipping. Stained tissue sections were imaged using a Leica Aperio AT2 Scanscope (Leica Biosystems, Deer Park, IL, USA).

### 2.9. Statistics

All data were assessed using Graph Pad Prism (Version 8.4.3). Differences in tumor incidence, tumor multiplicity, and tumor burden were compared between mammary cancer cell lines using one-way analysis of variance (ANOVA) with post-hoc comparisons performed. Data were considered significant when *p* < 0.05.

## 3. Results

To develop an isogenic, immune-competent mouse model of early-stage breast cancer progression, we used a previously established MIND model developed by our group [20,37], which allows for cancer cell lines to be injected directly up the teat of mice in the absence of surgical manipulation and surgery’s associated inflammation. Using this method, we characterized tumor formation and progression of six murine mammary tumor cell lines (D2A1, D2.OR, 4T1, EMT6, Py230, EO771) in two strains of immune-competent mice (BALB/c, C57BL/6) (Figure 1A).

### 3.1. Intraductal Injection of EO771 Murine Mammary Tumor Cells Results in Cell Dispersion without Compromising Ductal Integrity

We first confirmed that intraductal injection results in uniform dispersion of mammary cancer cells throughout the mammary gland. We assessed the spread of intraductally injected trypan blue, which displayed widespread intraductal dispersion throughout the mammary ductal tree (Figure 1B). Ductal integrity was inferred by the lack of leakage of trypan blue into the surrounding parenchyma (Figure 1B). We next assessed for dispersion and intraductal containment of tumor cells following intraductal delivery. EO771 tumor cells (1 × 10^5^ cells in 5 µL) were intraductally injected into the mammary gland of C57BL/6 mice and H&E analysis was performed on mammary glands collected 5 min post-tumor cell injection. Tumor cells were broadly dispersed within the ducts, with no evidence of tumor cell leakage into the stroma (Figure 1C). Since progression to invasive disease involves disruption of ductal integrity over time, we verified that intraductal delivery of tumor cells at the time of tumor cell delivery did not compromise the epithelial nor myoepithelial cell layer integrity. IHC analysis was performed to assess expression of E-cadherin and calponin as biomarkers of epithelial and myoepithelial cell integrity, respectively. No evidence of ductal disruption was observed (Figure 1D). All subsequent intraductal tumor cell injections were limited to 2 µL to further assure ductal integrity, a key requisite for the study of early-stage disease progression.

### 3.2. Murine Mammary Cell Lines Delivered Intraductally Model Human Invasive Breast Cancer

We next sought to compare the histology of mouse mammary tumors derived from mammary fat pad injections, the most common transplant model approach, to the histology of tumors derived from intraductal injection. Murine mammary cancer cell lines, Py230 or EO771, were intraductally injected into the 4th right mammary teat of C57BL/6 mice, using cell numbers comparable to previous MIND studies that used human breast cancer cell lines at 2 × 10^4^ to 5 × 10^4^ cells/injection [20,25,27]. In the contralateral 4th left mammary gland, the same number of Py230 or EO771 tumor cells were injected directly into the mammary fat pad. Tumors were collected 14 days (EO771) or 37 days (Py230) post-injection for histologic assessment.

Injection of EO771 (Figure 2A) and Py230 (Figure 2B) cells directly into the mammary fat pad resulted in the formation of solid, invasive tumors with necrotic cores. In contrast, intraductal injection of these same cells resulted in the development of invasive tumors with heterogeneous histology. Specifically, MIND EO771 tumors were characterized by an infiltrating fat phenotype (Figure 2C), similar to some human breast cancers (Figure 2E). MIND Py230 tumors were characterized as having a phenotype more comparable to ‘pushing’ border human breast cancers (Figure 2D,F). These data are consistent with previously published work demonstrating that tumors that arise from intraductal human breast cancer cell delivery more closely reflect human breast cancer histology than tumors derived from mammary fat pad injections [26,27,31]. 

### 3.3. Intraductal Injection of Murine Mammary Cell Lines into BALB/c Immune-Competent Hosts Results in IDC in the Absence of DCIS

We next assessed for in situ disease following intraductal delivery of murine mammary tumor cells. BALB/c mammary tumor cell lines, D2A1, D2.OR, 4T1, or EMT6, were intraductally delivered into the mammary gland of immune-competent BALB/c mice, at 1 × 10^4^ and 5 × 10^4^ cells per gland, cell numbers comparable to previous MIND studies using human breast cancer lines [20,25,27,29]. Mammary glands were collected between 11–33 days post tumor cell injection (Appendix A), timepoints prior to the development of overt lesions as assessed by palpation, but concurrent with microlesions as confirmed by histological assessment (Figure 3).

To evaluate the presence of in situ disease, myoepithelial layer integrity was measured via staining for the myoepithelial markers p63, calponin, and SMA, which are clinically used to delineate DCIS from IDC. While adjacent, uninvolved ductal myoepithelial cells clearly stained continuously positive for SMA and calponin, we were unable to observe intraductal lesions consistent with DCIS. Rather, we observed multifocal, invasive lesions associated with focal loss of myoepithelial cell integrity as measured by loss of SMA, calponin (Figure 3A,B), and p63 (Appendix A) staining, more aligned with that of pure IDC. Two BALB/C mammary tumor cell lines (D2A1 and D2OR) formed only IDC. Two additional BALB/c mammary tumor lines (4T1 and EMT6), when injected at lower concentrations, still robustly formed IDC-like lesions, but also displayed DCIS-like lesions in approximately 10% of cases (Figure 3B; 1/10 and 1/11 injections, respectively, Figure 3C; Appendix A; Appendix A). An example of DCIS is presented in Appendix A. These findings suggest that these BALB/c mammary tumor cell lines, when delivered intraductally into immune-competent hosts, associate early on with focally compromised myoepithelium, and progress rapidly to invasive disease without robust evidence of a stable, DCIS-like precursor stage.

One mechanism that may promote development of pure IDC compared to IDC development following a DCIS stage is the host’s immune response, which may compromise myoepithelial integrity and accelerate the progression to invasive disease. Thus, we assessed for CD45 (common lymphocyte antigen) expression and found an accumulation of CD45+ cells around these intraductally derived tumors (Figure 3A); an observation consistent with immune recognition of the tumor, despite tumor cells being derived from the same BALB/c background.

### 3.4. Intraductal Injection of Murine Mammary Cell Lines into C57BL/6 Immune-Competent Hosts Results in IDC in the Absence of DCIS

We next investigated a second immune-competent mouse strain, the C57BL/6 strain, for its ability to support DCIS development. C57BL/6 mammary tumor cell lines Py230 (1 × 10^4^ and 5 × 10^4^ cells, 2 µL) and EO771 (5 × 10^3^ or 5 × 10^4^ cells, 2 µL) were injected intraductally into mammary glands of C57BL/6 mice. Mammary glands were collected 14 days (Py230) or 35 days (EO771) post-injection; timepoints prior to the development of overt lesions as assessed by palpation, but concurrent with microlesions as confirmed by histological assessment (Figure 4).

Similar to our observations in the BALB/c model, we found that intraductal tumor cell injection resulted in the rapid establishment of multifocal invasive lesions within the mammary gland (Figure 4A,B). Intraductal injection of EO771 cells resulted in only invasive tumors at both high and low cell numbers (Figure 4C,D). The vast majority of Py230 lesions were locally invasive (57/63; 90.5% IDC), with occurrences of DCIS-like lesions in 9.5% of cases (6/63 injections DCIS) (Figure 4C and Appendix A). No glands showed DCIS alone. Similar to the BALB/c model, when assessed for CD45 expression, there was an accumulation of CD45+ cells around the intraductally derived tumors (Figure 4A), consistent with tumor cells eliciting an immune response in this C57BL/6 intraductal mouse model.

### 3.5. Adaptive Immunity Does Not Drive Rapid Progression of IDC

The influx of CD45+ cells into intraductally derived tumors in both BALB/c and C57BL/6 mice, together with high incidence of pure IDC, suggests that a host immune response might promote the development of pure IDC in these isogenic, immune-competent mouse models. Indeed, interactions between immune cells and tumor cells are suggested to play an important role in the progression to invasive disease [21,38]. Furthermore, previous studies have demonstrated that human breast cancer cell lines injected intraductally into immunocompromised mice progress through a long-lived (months) DCIS stage prior to invasion [20,25,26]. Therefore, we next sought to determine whether intraductal injection of murine mammary cancer cell lines into immunocompromised hosts similarly permits progression through a pre-invasive, in situ stage (i.e., DCIS), or whether tumors still rapidly progress to IDC in the absence of a functional adaptive immune system.

First, using human breast cancer cell lines (HCC70, MCF-7, T47D), we confirmed DCIS-like lesion formation in the absence of IDC in SCID/C57BL/6 (SCID) mice when using our surgery-free intraductal delivery model. HCC70, MCF7, and T47D cell lines were intraductally injected into SCID mice at 5 × 10^4^ cells in 2 µL, and mammary glands collected 4 weeks post-injection. DCIS-like tumors, without evidence of IDC, were evident 4 weeks following intraductal injection of HCC70 (six mice, six of six injected glands), and MCF7 (four mice, three of four injected glands) cell lines. These tumors were confirmed to be in situ lesions through H&E analysis and by evidence of continuous calponin staining in the surrounding myoepithelium (Figure 5A,B). T47D cells failed to form tumors in our MIND model (four mice, zero of four injected glands). Next, murine mammary tumor cell lines Py230 and EO771 were injected into the mammary glands of SCID/C57BL/6 (SCID) mice. In parallel, tumor cells were injected into wild-type C57BL/6 (WT) mice as controls. Intraductal delivery of Py230 and EO771 cells into immunocompromised mice again resulted in the rapid development of IDC, as determined by the loss of myoepithelial markers SMA and calponin (Figure 5C–F) with no evidence of DCIS observed. Intriguingly, while tumor incidence did not differ between SCID and WT mice, tumor multiplicity and average tumor size per gland was significantly increased in immunocompromised mice (Appendix A). Thus, the attenuated adaptive immune response in SCID mice led to more successful establishment and growth of lesions, consistent with the observed role of adaptive anti-tumor immunity in controlling invasive disease in immune-intact mice. In sum, these results do not support an obligate role for adaptive immunity in promoting tumor escape and progression to IDC through a DCIS stage in these isogenic mouse mammary tumor models (Appendix A, Appendix A).

Alternatively, we hypothesized that intrinsic tumor factors may contribute to the low incidence of IDC + DCIS, opposed to pure IDC. To assess whether these mouse cell lines are intrinsically more aggressive than the characterized human breast cancer lines, tumor cell migration was assessed using a classic scratch-based wound healing assay. We found that human breast cancer cell lines were significantly less motile compared to mouse cell lines, independent of the cell lines hormone receptor status and intrinsic subtype (Figure 5G; Appendix A). Specifically, human tumor cells T47D and MCF7 (luminal A) migrated to the same extent as human HCC70 tumor cells (basal-like), with all human cell lines significantly less motile than murine luminal A (EMT6, Py230), luminal B (D2A1, EO771), and basal (D2OR) subtypes. Together, these findings suggest a tumor intrinsic biology in murine cell lines that may promote the rapid and focal loss of myoepithelial cell differentiation markers and progression to IDC in the absence of a DCIS-like stage.

## 4. Discussion

In women, pure IDC is an aggressive disease with poorer survival outcomes compared to IDC concurrent with DCIS [3,8,9,10]. Understanding the molecular mechanisms that underlie disease progression, in the presence and absence of a DCIS stage, is essential for improved prognostics and for tailoring treatments to individual women. Despite the need for characterizing progression from in situ to invasive disease in the context of a fully competent immune system, there are a lack of tractable, immuno-competent models available for studying early-stage disease progression. In this study, we show that intraductal injection of murine mammary carcinoma cells readily resulted in the rapid development of invasive breast cancer without the formation of detectable stable DCIS. Similarly, in immunocompromised mice, intraductal injection of murine mammary carcinoma cell lines preferentially formed IDC without evidence of DCIS. We suggest that these mouse models would be useful for the study of human IDC that develop in the absence of a DCIS precursor stage. It is intriguing to consider whether the murine breast cancer MIND models we have described here might also provide insight into the biology that control early to late-stage disease progression, such as in interval breast cancer—i.e., cancers that emerge after a non-suspicious mammogram but before the patient’s next scheduled screen [39,40,41,42]. Interval breast cancers are rapidly progressing cancers that have been proposed to bypass a durable DCIS stage. Since DCIS is readily detected by mammographic screening whereas interval breast cancer is, by definition, mammographically silent, we speculate that these murine models may help delineate between mammographically detectable and silent disease.

In our isogenic MIND model, progression to IDC was characterized by the focal loss of myoepithelial cell differentiation markers, suggesting that loss of myoepithelial integrity is a key event in the progression to invasion in pure IDC. While the factors that control the loss of the myoepithelium remain unclear, numerous studies have implicated the immune system as a possible mediator. Immune cell accumulation is observed in areas of myoepithelium destabilization and microinvasion [19,21]. Furthermore, immune cell infiltration is associated with high-risk pathological features of DCIS, including grade, histologic subtype, and recurrence [38,43,44,45,46,47], as well as microinvasion [21,48,49,50,51] and metastasis [52,53,54,55]. Together, these data suggest that early tumor cell recognition and immune response to tumors may facilitate the loss of myoepithelial cell differentiation and the development of invasive disease in both pure IDC and in IDC concurrent with DCIS. However, our findings show that intraductal injection of EO771 and Py230 cell lines into immunocompromised mice—mice with an impaired adaptive immune system—resulted in the rapid loss of myoepithelial differentiation markers and development of IDC in the absence of DCIS. These data demonstrate that an intact immune system is not the sole driving factor for loss of myoepithelial barrier function. However, while these data do not support a primary role for the immune system for these established murine mammary tumor cell lines, they also do not rule out a role for immune modulation in disease progression. Further research is needed to better understand interactions between tumor cell subtypes, myoepithelial cell states, and the immune interactions that modulate disease progression. Other factors, including intrinsic tumor biology, species-specific interactions between mouse tumor cells and the mouse myoepithelium, as well as different stromal elements, may favor the development of IDC in the absence of DCIS in these isogenic MIND models of early disease.

Between experiments, there was variability in tumor burden when using the same cell line and injection conditions. In one experiment, injection of Py230 into C57Bl/6 wildtype mice resulted in an average lesion area of 0.32 mm^3^ (14 days post-injection; Appendix A); while in a second experiment average lesion area was 1.25 mm^3^ (11 days post-injection; Appendix A). Differences in cell viability at the time of injection, and/or subtle variations in the user’s injection technique, might influence the number of cells that successfully establish within the mammary duct. We also observed an inverse correlation between the average tumor area and the number of cells injected, whereby increasing cell concentrations results in reduced tumor burden. Variability in cell density within the duct may influence cell viability, differentially promote immune responses, and/or result in different associations with the myoepithelium; all of which could influence tumor progression. Together, these observations highlight the inherent complexity and heterogeneity of tumor biology within the mammary ducts. Of note, similar tumor size heterogeneity is commonly reported in mammary fat pad injection models. While variability in tumor size is one component of the MIND model, our observations that MIND model tumors better reflect human breast cancer histology and heterogeneity suggests that these mouse models could provide a useful tool for studying IDC that develops in the absence of DCIS. Increasing the number of mice and mammary glands injected, as well as optimizing cell numbers for each cell line, might help account for variability between studies.

We suggest species-specific tumor intrinsic biology combined with extrinsic factors likely contribute to the differences in IDC formation observed between the mouse and human tumor cell lines. Firstly, we find the murine cell lines to be intrinsically more motile than human breast cancer cell lines, independent of intrinsic subtype. Secondly, species-specific crosstalk between tumor cells and the murine mammary microenvironment may exist, including the myoepithelium, compared to human tumor cells, and such species-specific crosstalk may facilitate the rapid loss of myoepithelium differentiation markers and progression to IDC in the absence of DCIS. Our study has not addressed this important consideration, and, overall, crosstalk between tumor cells and myoepithelium is poorly characterized. The use of heterotypic cell culture models of human or murine mammary tumor cells mixed with human or mouse mammary epithelium may yield insight.

To date, a major disadvantage of isogenic implantable tumor models in the mammary fat pad, when compared to autochthonous tumor models, is their inability to accurately replicate the tumor immune microenvironment and anti-tumor effector T-cell response. Isogenic tumor models are known to result in enhanced anti-tumor immune responses compared to autochthonous models [56,57]. While isogenic models have been a mainstay of cancer therapeutics research, they are typically more inflamed and less responsive to immune-modulating therapy and do not accurately represent the co-evolution of tumor and anti-tumor immunity observed in autochthonous models and natural cancer development [56,58]. In breast cancer specifically, it has been demonstrated that tumors developing spontaneously in the MMTV-PYMT model are significantly less infiltrated with both effector and regulatory T cells, as well as myeloid cells, compared to tumors implanted into the mammary fat pad, and these differences tend to be exacerbated through implantation of higher numbers of cells at the time of inoculation [58]. Isogenic implantable models can be made less responsive to immunomodulation and radiotherapy through blockade of antigen presentation or depletion of adaptive immunity at the time of tumor implantation [57]. Tumors developing in our isogenic MIND model demonstrate comparable kinetics and IDC development in both immune-competent and immunodeficient animals, suggesting a lack of immune-mediated control of disease progression following intraductal injection. We propose that orthotopic IDC established with a MIND model may more accurately reflect the chronic antigen exposure, slow kinetics, and low antigen burden of autochthonous models known to result in effector dysfunction [58,59], with the advantage of synchronous tumor development and isolated gland involvement. If this is true, the use of MIND models of breast cancer in novel therapeutics research may result in the development of immune-modulating therapies in murine models which are able to be more directly translated to patient populations. A limitation to our study is that we did not characterize the immune cell infiltrate associated with early disease, and, thus, this provocative, potential advantage of immune-competent, intraductal models requires further study.

Our findings may be perceived as contrasting with published studies that demonstrate intraductal delivery of murine mammary tumor cells progress through an in situ stage. In such studies, intraductal delivery of murine mammary carcinoma cell lines (4T1, Py230, Mvt-1) at concentrations consistent with our study, were reported to develop DCIS-like lesions [60,61,62]. However, like our studies, these lesions were transient, and detectable only up to two weeks post-injection [60,61,62], and the analyses of myoepithelial integrity were limited. Furthermore, the methodology of these studies differs from ours. Several of the studies used a Y-shaped incision to visualize the inguinal gland and/or surgically remove the mammary teat to expose the main lactiferous duct [60,62], whereas our study was performed in the absence of surgery to minimize inflammation as a potential confounder [20]. One study injected tumor cells intraductally in a 1:10 ratio with Matrigel [61], which is highly enriched in basement membrane proteins and thus may bypass early stages of tumor cell-myoepithelial cell interactions. Further research is needed to harmonize the results across these various MIND model studies.

## 5. Conclusions

Currently, the mechanisms that underlie early-stage breast cancer progression to invasive disease remain unclear. The murine breast cancer MIND models we have described here appear well suited for modelling human invasive breast cancer that does not progress through an in situ stage. Pure IDC is a more aggressive disease with an increased risk of metastasis, compared to IDC diagnosed concurrently with a DCIS component, and represents at least 20% of all breast cancer diagnoses [8,9,10]. Furthermore, these immune-competent mouse models may provide significant advantages over mammary fat pad xenografts for modelling invasive disease, as the arising tumors more accurately reflect the histologic heterogeneity of human breast cancer and potentially permit greater immuno-editing and immune exhaustion, which is a requisite for the development of immune-modulating therapies. In summary, further research into the potential relevance of these isogenic, intraductally derived mouse mammary tumors as models for pure IDC and interval breast cancer are warranted, as are new approaches to modeling DCIS in immune-competent murine models.

## Figures and Tables

**Figure 1 cancers-15-02257-f001:**
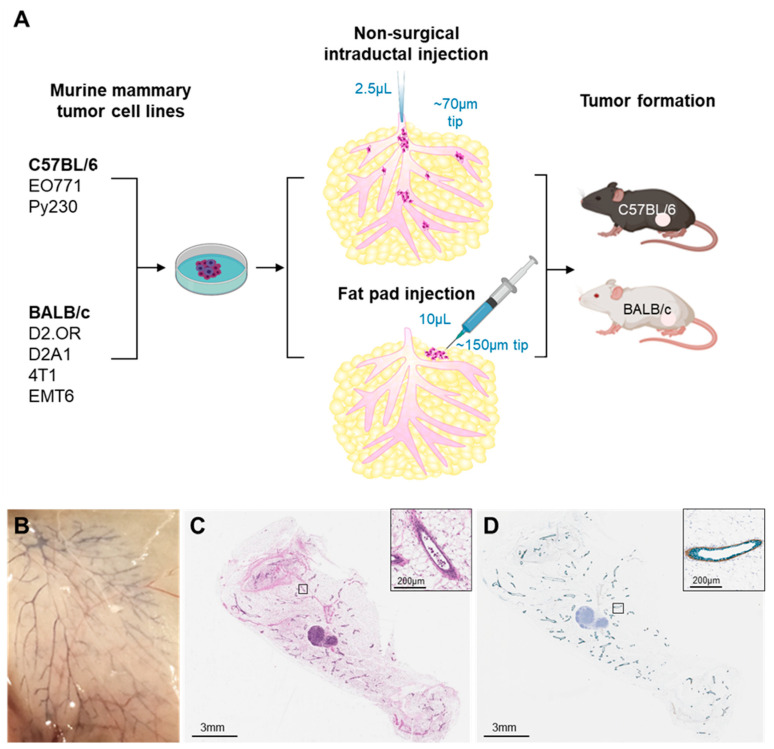
Validation of tumor cell dispersion and maintenance of ductal integrity after intraductal tumor cell delivery. (**A**) Schematic overview of the experimental procedure. (**B**) Dispersion of trypan blue dye (0.4% in PBS, 5 µL) throughout the mammary ductal tree following intraductal injection into the mammary gland of C57BL/6 mice. (**C**) Hematoxylin–eosin staining of mammary glands collected immediately after intraductal injection of EO771 cells (1 × 10^5^ cells in 5 µL). (**D**) Immunohistochemical analysis of E-cadherin (green) and calponin (brown) expression demonstrating maintained ductal and myoepithelial cell layer integrity. Insets show higher magnification.

**Figure 2 cancers-15-02257-f002:**
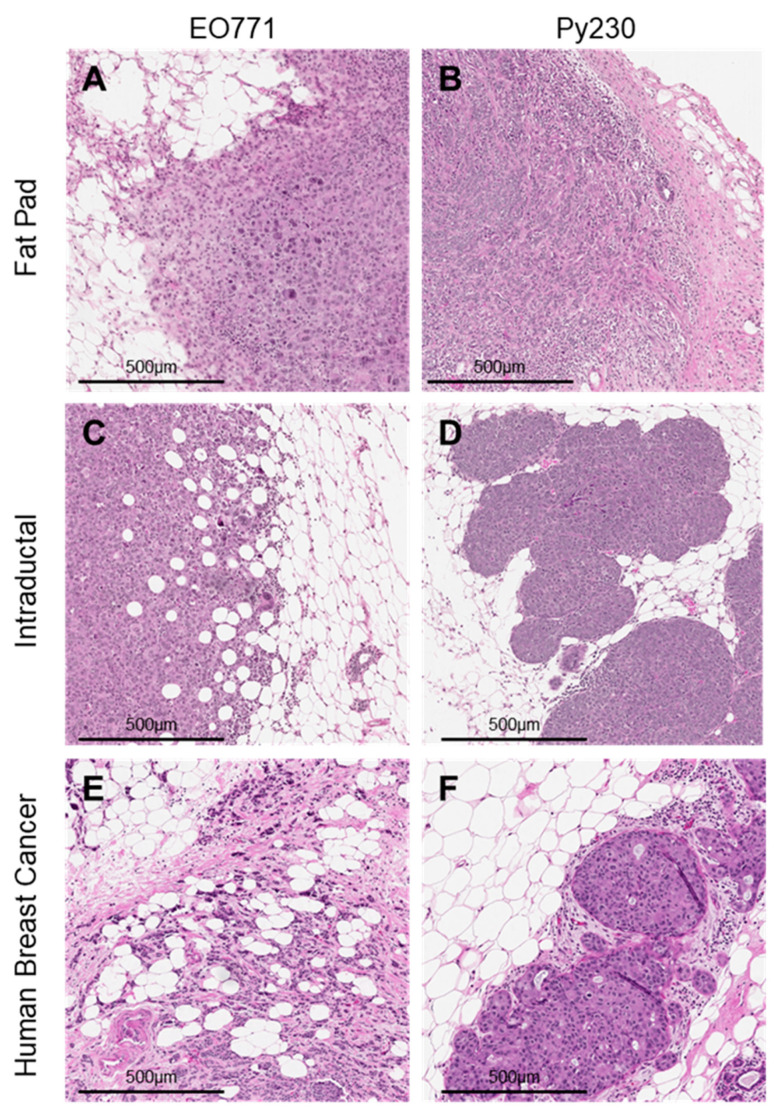
Histological analysis of tumors derived from intraductal injections, compared to mammary fat pad injections. Hematoxylin–eosin stains of tumors derived from the injection of murine mammary cell lines (**A**) EO771 (5 × 10^4^ cells in 10 µL) or (**B**) Py230 (5 × 10^4^ cells in 10 µL) into the 4th left mammary fat pad of C57BL/6 mice. In parallel, (**C**) EO771 (5 × 10^4^ cells in 2 µL) or (**D**) Py230 (5 × 10^4^ cells in 2 µL) were intraductally injected into the contralateral 4th right mammary gland. Hematoxylin–eosin stains of invasive human breast cancer samples characteristic of (**E**) infiltrating fat breast cancer and (**F**) ‘pushing’ border breast cancers.

**Figure 3 cancers-15-02257-f003:**
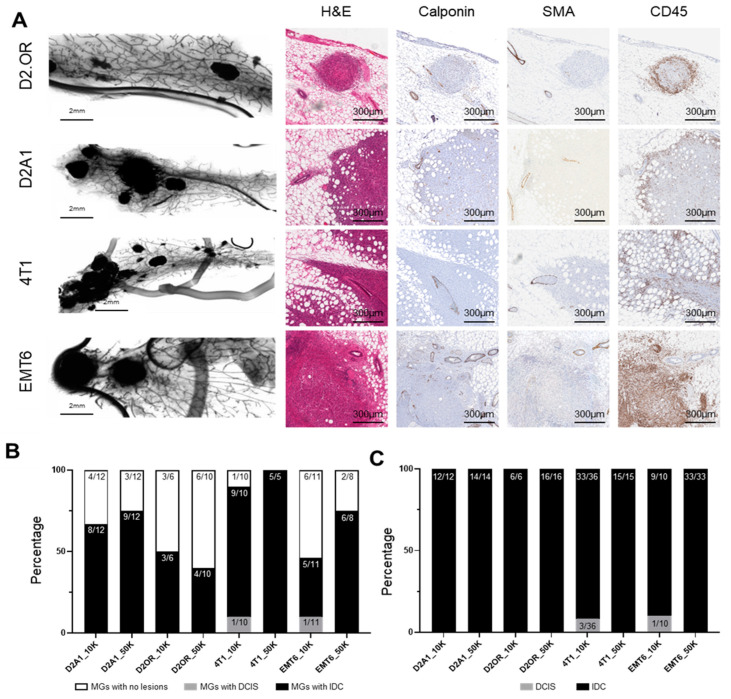
Characterization of tumors derived from intraductal injection of mammary cell lines into BALB/c immune-competent hosts. Murine mammary cell lines D2A1, D2.OR, 4T1, or EMT6 were intraductally injected into the 4th mammary glands of BALB/c mice at either 1 × 10^4^ (10K) cells or 5 × 10^4^ (50K) cells in 2 µL. Mammary glands were collected when the first tumor became palpable. (**A**) Whole mount and histochemical staining of mammary glands injected with murine mammary cancer cell lines. (**B**) The total number of injected mammary glands that developed tumors. (**C**) The proportion of lesions that were ductal carcinoma in situ (DCIS) or invasive carcinoma (IDC). Abbreviations: MG = Mammary Gland.

**Figure 4 cancers-15-02257-f004:**
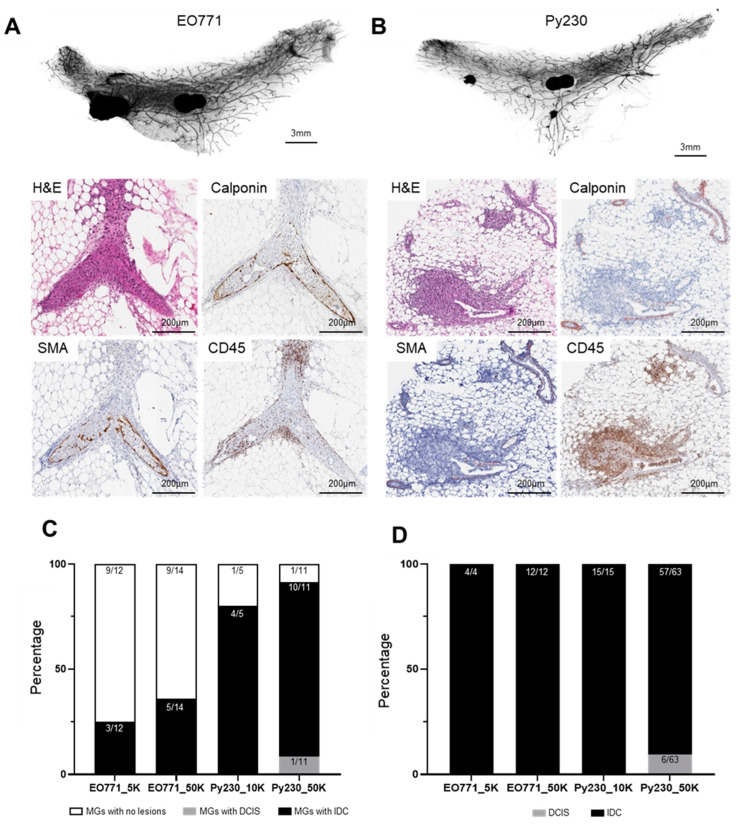
Characterization of tumors derived from intraductal injection of mammary cell lines into C57BL/6 immune-competent hosts. Murine mammary cancer cell lines EO771 or Py230 were intraductally injected into the 4th mammary glands of C57BL/6 mice at various concentrations: EO771 (5 × 10^3^ (5K) or 1 × 10^4^ (10K) cells in 2 µL) or Py230 (1 × 10^4^ (10K) or 5 × 10^4^ (50K) cells in 2 µL). Mammary glands were collected either 35 days (EO771) or 14 days (Py230) post-injection and assessed for tumors. Whole mount analysis and histochemical staining of tumors derived from intraductal injection of (**A**) EO771 and (**B**) Py230 cell lines. (**C**) The total number of injected mammary glands that developed tumors. (**D**) The proportion of lesions that were ductal carcinoma in situ (DCIS) or invasive carcinoma (IDC). Abbreviations: MG = Mammary Gland.

**Figure 5 cancers-15-02257-f005:**
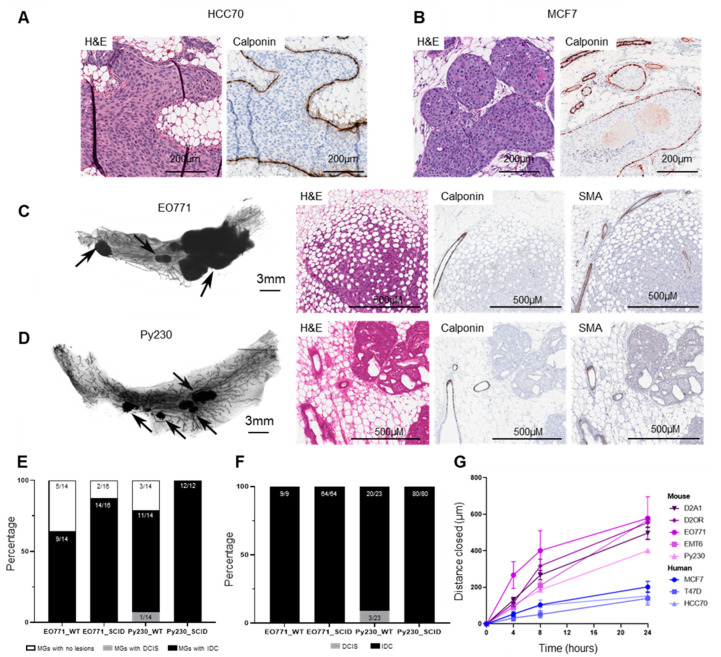
Characterization of tumors derived from intraductal injection of human breast cancer and mammary cancer cell lines into C57BL/6/SCID immunocompromised hosts. Representative hematoxylin–eosin stain and immunohistochemical analysis of calponin in mouse mammary glands injected with (**A**) HCC70 or (**B**) MCF7 breast tumor cells (5 × 10^4^ cells in 2 µL). (**C, D**) Murine mammary cell lines EO771 (5 × 10^4^ cells in 2 µL) or Py230 (5 × 10^4^ cells in 2 µL) were intraductally injected into the 4th right mammary glands of C57BL/6/SCID (SCID) mice or wild-type C57Bl/6 (WT) mice. Mammary glands were collected 26 days (EO771) or 11 days (Py230) later and assessed for tumors. Whole mount and histochemical staining of mammary glands collected from SCID mice that were injected intraductally with (**C**) EO771 or (**D**) Py230 mammary cancer cells. Tumors are identified by arrows. (**E**) The number of injected mammary glands that developed tumors following intraductal injection. (**F**) Proportion of lesions that were in situ (DCIS) or invasive carcinoma (IDC), as defined through p63, SMA, and calponin immunohistochemical analysis. (**G**) Distance closed by murine (D2A1, D2OR, EO771, EMT6, Py230; *n* = 5 experiments) and human breast cancer cell lines (MCF7 and T47D (*n* = 4), HCC70 (*n* = 3) during a 24 h scratch assay. Measurements were taken at 0, 4, 8, and 24 h post-scratch. Data presented as mean ± standard error of the mean. Abbreviations: MG = Mammary Glands.

## Data Availability

The datasets used and/or analyzed during the current study are available from the corresponding author on reasonable request.

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
