# Peer review of "Isogenic Mammary Models of Intraductal Carcinoma Reveal Progression to Invasiveness in the Absence of a Non-Obligatory In Situ Stage"

_cancers, 2023, doi:10.3390/cancers15082257_

Round 1
Reviewer 1 Report
I am really grateful for reviewing this manuscript. In my opinion, this manuscript can be published once some revision is done successfully. In this study, intraductal injection of murine mammary carcinoma cells readily resulted in the rapid development of invasive breast cancer without the formation of detectable stable DCIS (ductal carcinoma in situ). Similarly, in immunocompromised mice, intraductal injection of murine mammary carcinoma cell lines preferentially formed IDC (invasive ductal carcinoma) without evidence of DCIS. The authors suggest that these mouse models would be useful for the study of human IDC, which develop in the absence of a DCIS precursor stage. I would argue that this is a rare achievement. However, it needs to be noted that different animal experiments would involve different control variables (e.g., different procedures). I would like to ask the authors to address this issue in greater detail in the section of Discussion.
Reviewer 2 Report
In this manuscript, authors take advantage of pre-established intraductal tumor cell delivery model and combine it with isogenic murine mammary carcinoma cell lines (D2A1, D2.OR, 4T1, EMT6, EO771, and Py230) and immune-competent hosts (BALB/c and C57BL/6) to model human invasive ductal carcinoma (IDC) that bypasses ductal carcinoma in situ (DCIS). In addition, they use one immunocompromised (SCID) mouse strain and three human cancer cell lines (HCC70, MCF7, and T47D) to demonstrate that the immune system alone is not the sole driving factor for rapid disease progression. The authors confirmed that intraductal injection leads to uniform dispersion of mammary cancer cells throughout the mammary gland. Additionally, they claimed that tumors arising from intraductal delivery more accurately reflect the histology of human breast cancer than tumors derived from mammary fat pad injections.
The authors focused on assessing for the presence of in situ disease after Intraductal injection of murine mammary cell lines into immune-competent hosts, by using the myoepithelial markers and which results in IDC in the absence of DCIS. The hypothesis that a host immune response could promote the development of pure IDC in immunocompetent isogenic models was explored, and the effect of intraductal injection of murine mammary cancer cell lines into immunocompromised hosts was studied. Human mammary cancer cells were used as a control for DCIS-like tumors, which did not show evidence of IDC. The authors confirmed that intraductal delivery of Py230 and EO771 cell lines into immunocompromised mice resulted in the rapid development of IDC without evidence of DCIS. To evaluate whether intrinsic tumor factors may contribute to the rapid progression to pure IDC, the authors performed a wound healing assay to assess the aggressiveness of mouse and human cell lines, demonstrating that human cancer cells were less motile compared to mouse cells, regardless of their status or subtype.
The findings presented in this study are intriguing as they shed light on potential ways to replicate the development of IDC in tractable immunocompetent animal models to enhance our understanding of the underlying mechanisms of disease progression, with or without a DCIS stage. The authors provide strong evidence to support their conclusions, and the data provided are of high quality, with appropriate controls. However, some of the conclusions drawn from the data may be limited due to a superficial analysis, and the mechanistic insights are limited. Additionally, certain aspects of the data interpretation need to be addressed to further improve the study. It is also worth noting that some control experiments aimed at validating the author's conclusions are missing, which could have strengthened the findings.
The following are some questions to address:
1. Supplementary Figure 2, which is not mentioned in the text, and which contributes quantitatively to Figure 4, shows a significant inverse correlation between the average area of lesions and the number of cells injected. It is indeed contradictory that the more cells injected, the smaller the resulting lesion area. However, it is possible that this phenomenon may be due to the complex interactions between the injected cells, the host microenvironment, and the immune system, which can have a significant impact on the development and growth of lesions. The authors should provide a more detailed explanation for this phenomenon in their discussion.
2. Regarding the experiment of Figure 5 (in WT mice) in which they use the same isogenic model as figure 4, the average area of developed lesions (1.25 mm2) is also larger even after fewer days post injection (11 days) than the previous experiment in Figure 4 (0.322 mm2,14 days). Concerning the variability in the data, it is important for the authors to acknowledge and address this issue. The authors should highlight the limitations of their study and provide suggestions for future research to address these limitations.
3. Figure 5G: Invasion assays (trans-well) rather than migration could help to explain better the difference in the results obtained with human and murine cells.
4. The authors suggest that intrinsic tumor biology, species-specific interactions, and variations in stromal elements may contribute to the development of IDC in their MIND models, even in the absence of DCIS. To further elucidate the molecular mechanisms underlying their model, it would be of great interest to include assays that assess the secretion of protease inhibitors such as Stefin A, SERPINI2, or TIMP-1 by myoepithelial cells, which are known to exert an inhibitory effect on cancer growth and invasion. These assays could help to clarify the role of myoepithelial cells in the rapid development of pure IDC and provide further insights into the complex interactions between tumor cells and the microenvironment.
Other minor points:
I. All the supplementary figures and tables should be mentioned throughout the article, and discussed when they add valuable information.
II. Stains for the p63 marker should be in the main figures or mention their locations on the main text (Supplementary Figure 4). Magnifications with arrows or arrowheads showing focal loss of myoepithelial cell integrity would be helpful.
III. Legend of Supplementary Figure 2 has an error: “concentrations: EO771 (5×103 (5K) or 1×104 (10K) cells in 2μL)”, please correct it.
IV. Error at line 303 in the statement “Mammary glands were collected between 11-33 days post tumor collection”. Do you mean post-injection?
V. Please clarify this sentence in the legend of Figure 5 and Supplementary Figure 3 “Mammary glands were collected 2 weeks later and assessed for tumors” which is contradicted in Supplementary Table 3 (days of euthanasia 26 or 11).
Reviewer 3 Report
Overall, the article provides interesting insights into the use of isogenic murine models to study IDC. However, there are some points that need to be addressed.
1.The authors reported about 10% mouse occurrences of DCIS-like lesions in their mouse model, which they proposed as a tool to study IDC in the absence of a non-obligatory DCIS stage. However, this statement is not entirely appropriate as 10% is not a rare percentage.
2. All six murine mammary tumor cell lines injected in the study are invasive, with 4T1 and EO771 known for their highly metastatic behavior. It is unclear why these aggressive cell lines were chosen to study the progression of IDC. If the focus is on the early stages of breast cancer, it would be better to use transgenic mouse models such as the MMTV-ErbB2 mouse model for DCIS-like lesions that progress to invasive breast cancer. The authors also noted that human breast cancer cell lines injected intraductally into immunocompromised mice progress through a long-lived DCIS stage prior to invasion, suggesting that a human cell model may more accurately reflect the mechanisms underlying early-stage breast cancer progression to invasive disease.
3. The authors did not find any DCIS in their D2.OR mouse model, which differs from other studies that have reported the development of DCIS-like lesions using the same model. The authors should provide possible reasons for this difference.
4. To better understand the intrinsic biology of these cell lines, the authors should consider adding an invasion assay to their in vitro experiments.
5.If the tumor intrinsic biology of murine cell lines promotes rapid progression to IDC in the absence of a DCIS-like stage, it means that murine cell lines cannot form DCIS regardless of whether they are in wild-type or immunocompromised mice. This result makes it unreasonable to assume an obligate role for adaptive immunity in promoting tumor escape and progression to IDC through a DCIS stage.
6.The authors mention the use of human breast cancer cell lines injected intraductally into immunocompromised mice to study early stage breast cancer progression. It would be helpful if they could provide more data on this approach, such as the number of injected mammary glands that developed tumors and the proportion of lesions that were in situ (DCIS) or invasive carcinoma (IDC).
Round 2
Reviewer 3 Report
The author has properly responded all questions.